# Human Endocrine-Disrupting Effects of Phthalate Esters through Adverse Outcome Pathways: A Comprehensive Mechanism Analysis

**DOI:** 10.3390/ijms241713548

**Published:** 2023-08-31

**Authors:** Yunxiang Li, Hao Yang, Wei He, Yu Li

**Affiliations:** College of Environmental Science and Engineering, North China Electric Power University, Beijing 102206, China; 120222232055@ncepu.edu.cn (Y.L.); 120212232080@ncepu.edu.cn (H.Y.); 120222132018@ncepu.edu.cn (W.H.)

**Keywords:** phthalate esters (PAEs), adverse outcome pathway (AOP), machine learning, endocrine disrupting effects

## Abstract

Phthalate esters (PAEs) are widely exposed in the environment as plasticizers in plastics, and they have been found to cause significant environmental and health hazards, especially in terms of endocrine disruption in humans. In order to investigate the processes underlying the endocrine disruption effects of PAEs, three machine learning techniques were used in this study to build an adverse outcome pathway (AOP) for those effects on people. According to the results of the three machine learning techniques, the random forest and XGBoost models performed well in terms of prediction. Subsequently, sensitivity analysis was conducted to identify the initial events, key events, and key features influencing the endocrine disruption effects of PAEs on humans. Key features, such as Mol.Wt, Q^+^, QH^+^, E_LUMO_, minHCsats, MEDC-33, and EG, were found to be closely related to the molecular structure. Therefore, a 3D-QSAR model for PAEs was constructed, and, based on the three-dimensional potential energy surface information, it was discovered that the hydrophobic, steric, and electrostatic fields of PAEs significantly influence their endocrine disruption effects on humans. Lastly, an analysis of the contributions of amino acid residues and binding energy (BE) was performed, identifying and confirming that hydrogen bonding, hydrophobic interactions, and van der Waals forces are important factors affecting the AOP of PAEs’ molecular endocrine disruption effects. This study defined and constructed a comprehensive AOP for the endocrine disruption effects of PAEs on humans and developed a method based on theoretical simulation to characterize the AOP, providing theoretical guidance for studying the mechanisms of toxicity caused by other pollutants.

## 1. Introduction

Phthalate esters (PAEs) are phthalic acid derivatives with a characteristic odor. They are insoluble in water but soluble in most organic solvents [1]. PAEs are mainly found in five types of plastics: PA, PP, PVC, PET and PE [2]. PAEs can improve the flexibility of plastics and act as plasticizers [3,4]. They were used in products such as toys and food packaging materials, which caused serious environmental hazards and attracted global attention [5]. In 1999, the European Union issued a directive (1999/815/EEC) to strictly limit the content of six plasticizers (DEHP, DBP, BBP, DINP, DIDP, and DNOP) (<0.1%) [6]. China has also taken action against the environmental pollution caused by PAEs to mitigate the threat to human health. In 1995, China included three PAEs (DMP, DBP, and DOP) on the blacklist of priority pollutants [7]. In 2008, China stipulated that 16 PAEs should not exceed 0.05 mg/kg in food plastic packaging materials [8].

Phthalate esters (PAEs) can enter the body through breathing, eating, and skin contact, and participate in a series of toxic reactions, which in turn adversely affect the reproduction and development of the organism [9,10]. PAEs can bind to various hormone proteins in the body and exert different types of biological endocrine disrupting (ED) effects by acting as xenoestrogens [11], anti-androgens [11], and anti-thyroid hormones [12]. Raha et al. found that PAEs can competitively bind to natural estrogens and have a strong binding ability to estrogen receptors [13]. Hashemipour, et al. [14] found that the concentration of phthalate exposure was positively correlated with the incidence of precocious puberty in girls, which confirmed that PAEs had intrinsic estrogenic activity and endogenous ED effects. Josh et al. compared the binding affinity of PAEs to androgen receptors with corresponding natural steroids and known endocrine-damaging heterologous bisphenol A (BPA), and found that PAEs had a stronger binding affinity to androgen receptors [15]. Li, et al. [16] found that PAEs can downregulate androgen synthesis genes and inhibit the expression of insulin-like factor 3 (Insl3), resulting in male testicular hypoplasia. Sugiyama, et al. [17] showed that PAEs can exhibit the same effect as thyroxine after entering the human body; they compete with thyroxine and affect the binding of normal thyroxine to proteins, thus inhibiting the expression of the TR-β gene [18]. Because PAEs have many adverse effects on the human body by disrupting hormone signal transduction, and Santiago, et al. [19] found that THβ1 is responsible for the feedback regulation of the hypothalamus–pituitary–thyroid axis, the thyroid hormone receptor in this paper chooses TRβ1 instead of TRα1.

Adverse outcome pathways (AOPs) are used to describe the toxic pathways and modes of action of exogenous chemicals [20]. They consist of molecular initial events (MIEs), key events (KEs), and adverse endings (AEs) that lead to acute toxicity, cancer, and developmental and reproductive diseases [21]. AOPs are constructed to elucidate the reaction mechanism of chemicals. Brix, et al. [22] identified a variety of potential mechanisms of nickel toxicity and its interaction with freshwater aquatic organisms by constructing the AOP of nickel toxicity to aquatic organisms. Li, et al. [23] constructed the AOP of the adverse effects of environmental particulate matter on various systems and identified the mechanism by which environmental particulate matter triggers toxic effects on various systems. Liu, et al. [24] determined the specific mechanism of carcinogenesis by constructing an AOP of human bladder cancer induced by aromatic amines, and screened out aromatic amine antioxidant substitute molecules with low carcinogenicity of bladder cancer. Most researchers use 2D-QSAR models to construct AOPs; machine learning models are rarely used in such studies. The 2D-QSAR model is most often constructed using the MLR method, which has certain limitations. Many types of machine learning model and algorithms are available, and thus, there are many ways to solve regression problems. Here, we constructed a QSAR model based on a machine learning model to provide theoretical support for research on pollutant toxicity.

The binding of PAEs to proteins influences binding to DNA elements, which in turn affects the transcription and translation of target genes, interferes with the secretion of hormones in humans, and thus causes adverse reactions. In this study, an AOP was constructed to determine the molecular ED effects of PAEs. The binding of PAEs to an estrogen protein, androgen protein, and thyroid hormone protein was considered to be the molecular initial event (MIE). The action of the complex formed by PAEs and three hormone proteins on the specific sequence of DNA response elements, which induced or inhibited the transcription and translation of subsequent target genes, was considered to be the key events (KEs). The ED effect of PAEs on humans was considered to be the adverse ending (AE) when studying the mechanism of the ED effects of PAEs on humans. We also identified the key features affecting the AOP of the whole process of PAEs’ molecular endocrine-disrupting effect. In this study, (1) a machine learning model (MLR, RF, and XGboost model) for the AOP was constructed. A sensitivity analysis was performed to identify the key characteristics affecting the AOP; (2) a 3D-QSAR model of PAEs was constructed, and the relationship between the molecular structural characteristics of PAEs and their environmental effects was analyzed; and (3) based on molecular docking, the identity of amino acid residues, the contribution of BE, and various non-bonding forces were analyzed to elucidate the molecular mechanism of the ED effects of PAEs associated with AOPs. Our findings provide new information and theoretical guidance for studying the molecular mechanism of the ED effects of emerging pollutants such as PAEs.

## 2. Results and Discussion

### 2.1. Calculation of the Combined Effect of the Initial and Key Events in the AOP Associated with the ED Effects of PAEs

#### 2.1.1. Calculation of the Comprehensive Effect Value of the Initial Event in the AOP Associated with the ED Effects of PAEs

In this study, the BE of 22 PAEs to ER, AR, and THR proteins was calculated using molecular docking and MD simulations. The degree of interaction between PAEs and hormone protein receptors was expressed by BE. A smaller BE indicates a stronger binding ability and a greater hormone effect [25]. To more intuitively analyze the ED effects of PAEs and determine the main molecular characteristic values that have a greater impact on their ED effects, the CV of the BE data of the molecular docking between the 22 PAEs and the estrogen protein, androgen protein, and thyroid hormone protein was calculated using the CV method; the CVs were 0.5106, 0.4583, and 0.4458, and the weights were 0.3609, 0.3240, and 0.3151, respectively. PAEs had a strong ED effect on organisms, and the impact of the estrogen effect was more prominent. One study showed that the estrogen effect is slightly higher than the androgen effect and the thyroid hormone effect [26], which is consistent with the weight calculated in this study. The weight was calculated using the CV method, and the comprehensive effect value of the ED effects of PAEs was calculated, as shown in Table 1.

#### 2.1.2. Calculating the Comprehensive Effect Value of the Key Events in the AOP Associated with the ED Effect of PAEs

In this study, using the docking function module of HDOCK SERVER (http://hdock.phys.hust.edu.cn/ (accessed on 8 December 2022)), we defined the DNA response element as the ligand and the PAE–hormone protein complex as the receptor. The DNA response element score of the PAE–protein complex was calculated [27]. The data with the smallest RMSD was selected as the score, and the absolute value of the score was used to characterize the intensity of the effect of PAEs on the three hormone DNA response elements (the greater the absolute value of the score, the greater the effect of PAEs on the DNA of the three hormones). To determine the main molecular characteristic values of the effects of PAEs on the DNA of the three hormones, multiple correlation coefficients of the three groups of scores of estrogen DNA elements, androgen DNA elements, and thyroid hormone DNA elements were calculated. The correlation coefficients were 0.277, 0.295, and 0.110, and the weights were 0.224, 0.211, and 0.565, respectively. Therefore, compared to the effects of estrogen and androgen, the thyroid hormone effect of PAEs at the DNA level was more significant, which matched the results of a study by Shen et al., who investigated the in vitro hormone activity of PAEs using reporter gene analysis [28]. Thus, the weight of the hormone effect calculated by the multiple correlation coefficient method was reasonable. The comprehensive data of DNA response element scores are presented in Table 2.

### 2.2. Dimensionality Reduction on Descriptors of SMs and SMs Derivatives Using the Pearson Correlation Coefficient Method

In this study, by adjusting the classification threshold of the PCC model to 0.75, the eigenvalues with large correlations were removed independently, and the key eigenvalues with small correlations were left. In total, 19 main characteristic parameters with correlations within a reasonable range were screened (Figure 1). The process included importing the code package, reading the dataset, followed by PCC analysis; classification, screening, and dimension reduction; generating Pearson’s heat map.

### 2.3. Construction of a Machine Learning Model for the AOP Associated with the ED Effects of PAEs

#### 2.3.1. Construction of a Machine Learning Model for the Initial Event in the AOP Associated with the ED Effects of PAEs

To determine the mechanism of action of PAEs on the three hormone proteins, 19 characteristic values of PAEs screened with the PCC method were used as independent variables, and the comprehensive value of the absolute value of the BE of 22 PAEs and hormone proteins was used as the dependent variable. Python software was used to construct the MLR, RF, and XGboost models of the AOP initiation event of the ED effects of PAEs. The R^2^ values of the RF and XGboost model were 0.852 and 0.859 (>0.75), respectively [29], indicating that the RF and XGboost regression model both had a good fitting ability; the R^2^ of the MLR model was always negative, indicating that its performance was poor. Therefore, the RF and XGboost models of the initial event of the ED effect of PAEs were screened as qualified models.

#### 2.3.2. Construction of a Machine Learning Model for the Key Events in the AOP Associated with the ED Effects of PAEs

To determine the mechanism of the interference effect of PAEs on DNA response elements, the eigenvalues of 19 PAEs screened by the PCC method were used as independent variables, and the comprehensive values of the scores of DNA response elements of the three hormone proteins were used as the dependent variables. The MLR, RF, and XGboost models of the key events in the AOP associated with the ED effects of PAEs were constructed using Python software. The maximum R^2^ values of the RF regression model and the XGboost regression model were 0.855 and 0.836 (>0.75), respectively, indicating that the RF and XGboost regression models had a good fitting ability. The R^2^ of the MLR model was always negative, indicating that its performance was poor. Therefore, the RF and XGboost models of the KEs of the ED effects of PAEs were considered to be qualified models, and the key influencing factors of the ED effects of PAEs associated with the AOP were further screened.

#### 2.3.3. Construction of a Machine Learning Model for the Adverse Ending in the AOP Associated with the ED Effects of PAEs

In the initial event of the ED effects of PAEs, the binding energies of 22 PAEs with the estrogen protein, androgen protein, and thyroid hormone protein complexes were calculated using molecular docking technology and MD, and the comprehensive value of the ED effect of PAEs was calculated by the CV method; the HDOCK SERVER (http://hdock.phys.hust.edu.cn/ (accessed on 8 December 2022)) was used to calculate the scores of the 22 PAEs and estrogen, androgen, and thyroid hormone DNA response elements in the key events of the ED effects of PAEs. The comprehensive values of PAEs and molecular hormone DNA response elements were calculated using the multiple correlation coefficient method. To screen the key eigenvalues of the combined effect of the PAE hormones on DNA response elements, we used the forward data processing method to normalize the calculated comprehensive values of PAEs and receptor proteins and their corresponding DNA response elements. The entropy method was used to calculate the weight ratio of the initial event to the key event as 0.579:0.421 to obtain the harmful outcome effect value of the ED effects of PAEs (details in Table 3). Adam and Mhaouty-Kodja [30] analyzed the interaction between protein and DNA after exposure to PAEs and found that proteins strongly influenced the effect of PAEs on human testosterone, which was consistent with the weight of the initial event and the key event of the harmful outcome path of the ED effects of PAEs found in this study.

We also used the 19 eigenvalues of PAEs as independent variables, and the risk effect values of the harmful outcome pathways of the ED effects of the 22 PAEs were used as dependent variables. Python software was used to construct the RF and XGboost regression models of harmful outcomes in the AOP associated with the ED effects of PAEs. The R^2^ of the RF regression model was 0.751 (>0.75). The R^2^ of the XGboost regression model was 0.948 (>0.75), which indicates that the XGboost regression model had good predictive ability. Although the performance of the RF model was slightly worse than that of the XGboost model, the model had satisfactory applicability, rationality, and predictive ability.

### 2.4. Identification and Analysis of the Key Eigenvalues of the AOP Associated with the ED Effects of PAEs Based on the Sensitivity Analysis Method

In this study, sensitivity analysis was performed to analyze three RF model parameters of the initiation event, key event, and adverse ending of the AOP associated with the ED effects of PAEs. Among them, the key eigenvalues that significantly affected the initial event were Mol.Wt, Raman, and E_LUMO_; the key eigenvalue that significantly affected the key event was Q^+^; and the key eigenvalues that significantly affected the harmful outcome were Mol.Wt and Q^+^. The key eigenvalues that significantly affected the harmful outcome appeared in the initial event and the key event, respectively. The parameter Mol.Wt was the molecular weight of PAEs, and Q^+^ was the most positive Millikan charge number of the PAEs [31,32]. The results of the sensitivity analysis showed that, with the increase in Mol.Wt and Q^+^, the risk effect value of the harmful outcome path of PAEs also increased, which might produce stronger ED effects on the human body. Among them, Q^+^ was the most significant key eigenvalue affecting the key events, and its average SC was only 0.008. With the increase in Q^+^, the score of the DNA response element did not change significantly. The average SC of Mol.Wt in the initial event was 0.342, and the value of BE changed significantly with the increase in Mol.Wt. We concluded that in the harmful outcome pathway of the ED effects of PAEs, the initial event played a major role and the key event played a secondary role. This conclusion was consistent with the results reported by Li, et al. [33], who investigated the risk of artificial musk molecules causing abortions in pregnant women and showed that the binding ability of artificial musk molecules to proteins strongly promotes the harmful outcome pathway.

A sensitivity analysis (Appendix A) was conducted on the three XGboost models of the initial event, key event, and harmful outcome of the ED effect of PAEs related to the AOP. The key eigenvalues affecting the initial event were Mol.Wt, QH^+^, and MDEC-33, the key eigenvalues affecting the key event were Mol.Wt and Raman, and the key eigenvalues affecting the harmful outcome were Mol.Wt, Q^+^, QH^+^, E_LUMO_, minHCsats, MDEC-33, and EG. The eigenvalue Mol.Wt appeared in the initial event and the key event. QH^+^ and MDEC-33 were identified as the key eigenvalues in the initial event. Among them, QH^+^ represented the most positive Millikan hydrogen atom charge number of the PAEs, and MDEC-33 is the distance edge of PAEs and is a topological structure parameter [34], which is positively correlated with the risk effect value of harmful outcomes. In the three XGboost models, the average sensitivity coefficients of the three eigenvalues Mol.Wt, QH^+^, and MDEC-33 affecting the initial event model were 0.503, 0.422, and 0.307, respectively, while the average sensitivity coefficients of the eigenvalues Mol.Wt and Raman affecting the critical event model were 0.033 and 0.017, respectively. The score of the DNA response element did not change significantly with changes in Mol.Wt and Raman. This result confirmed that in the AOP associated with the ED effect of PAEs, the influence of the initial event on the degree of occurrence of the whole pathway was greater than that of the key event, which further verified the rationality of the weight of the initial event and the key event of AOP associated with the ED effect of PAEs calculated by the entropy method in this study.

By comparing the key eigenvalues of the RF and XGboost models of the AE related to the ED effects of PAEs, Mol.Wt and Q^+^ were found to be the key eigenvalues that affected the harmful outcome. Among them, molecular weight (Mol.Wt) was identified as a key eigenvalue in all five models, and the SC of Mol.Wt was almost the largest among the key eigenvalues of the five models. Thus, we inferred that Mol.Wt is the most critical factor affecting the ED effects of PAEs associated with AOP; additionally, it is often used to characterize the hydrophobic interactions between ligands and receptors [35]. Studies have shown that high-molecular-weight PAEs are dominant among the PAEs that people are exposed to every day, and the ED effects of high-molecular-weight PAEs on the human body are also stronger than those of low-molecular-weight PAEs [36]. When exposed to PAEs, PAE molecules can pass through the cell membrane to bind to hormone receptor proteins, and then, the molecule–protein complex binds to the specific DNA sequence in the corresponding DNA response element, inducing the transcription and translation of subsequent target genes and causing ED in the human body. Aromatic compounds, aliphatic hydrocarbons, and halogens are the hydrophobic parts of ligand PAEs. Hydrophobic contact is caused by the spatial proximity of the non-polar amino acid side chains and the hydrophobic substituents on the ligand PAE molecules. Water molecules are released from the hydrophobic region upon hydrophobic contact, and the unconstrained water molecules released can participate in the energy-favorable hydrogen bonding interactions, which enhance the overall binding affinity of the ligand [37,38,39]. Therefore, the hydrophobic interactions between ligands and receptors affect the ability of PAEs to bind to hormone proteins and influence the ability of PAEs to bind to DNA response elements. The enhancement of its binding ability represents the ED effects of PAEs. This increases the initiation events and key events, which strongly affect the synthesis, secretion, transportation, and binding of natural hormones in the body [40]. For example, because of the estrogenic characteristics of PAEs, they compete with the binding of estrogen to protein receptors in the body, disrupting the normal estrogen levels and disturbing hormone signal transduction in the body (e.g., the hypothalamus–pituitary–gonadal axis and hypothalamus–pituitary–thyroid axis), thus resulting in many adverse effects in the body [41]. Concerning the electronic parameters of ligand PAEs, Q^+^, QH^+^, and E_LUMO_ (the lowest occupied orbital energy) represent the electron gain and loss ability of PAEs. The binding of PAEs to hormone proteins to form a complex and the subsequent binding of the complex to DNA response elements can affect the hydrogen bonds and the electrostatic or polar interactions between them [42]. With the increase in Q^+^ and QH^+^, it is easier to form hydrogen bonds (HB) and VDW forces between ligands and receptors [43]. Thus, PAEs are more likely to bind to hormone proteins and DNA response elements due to the hydrophobic interactions caused by hydrogen bonds and van der Waals forces, which can enhance the ED effects on the human body. Additionally, the parameter E_LUMO_ related to the ionization potential of the compound can characterize the sensitivity of the compound to the electrophilic reagent. The higher the E_LUMO_, the easier it is for the compound to release electrons and the stronger the reactivity [44]; E_LUMO_ is also associated with molecular polarity and molecular electronegativity [45,46]. Changes in molecular polarity can affect the hydrophobic interactions between molecules and receptors, and enhancing the electronegativity of molecules promotes the formation of HB [47]. This can enhance the degree of interaction between PAEs and hormone proteins to increase the stability of binding, thus increasing the degree of initial events in the AOP associated with the ED effects of PAEs. These changes can increase the ED effects of PAEs on the human body.

The MDEC-33 represents the molecular distance edge and is a topological structure parameter related to the properties of molecular chemical bonds (such as number, type, and bond length) [34]. According to the BE of the three ED effects of PAEs calculated in this study, we found that the ED effects of PAEs on the human body were positively correlated with the number of chemical bonds of PAE molecules. Chain-like PAE molecules are more likely to bind thymine to the A-T base pair of DNA [48], which can inhibit the transcription of specific target genes and interfere with the endocrine system. Based on the results of the above analysis, we concluded that the molecular weight, HB, and VDW can significantly affect the degree of harmful outcome path of PAEs on the human body. We also confirmed the accuracy and reliability of the AOP RF model and the XGboost model for the ED effects of PAEs constructed in this study.

### 2.5. Validation of Key Characteristic Values of the ED Effects of PAEs Associated with the AOP

Some studies have shown that each round of iterative data of the XGboost model is related to the previous round of results, which ensures that the results are close to the real data analysis to a large extent [49]. The RF model is based on the Bagging method, each classification is independent, and different input data have negligible effects on the model [49]. The R^2^ of the RF model and the XGboost model for the harmful outcomes related to the ED effects of PAEs were 0.751 and 0.948, respectively (R^2^ is the percentage of the predictive variable that can explain the variation of the outcome variable; R^2^ values closer to 1 indicate that the model can better fit the data [50]). To summarize, the performance of the XGboost model was better and more accurate than that of the RF model, which was similar to the findings of a study by Hong, et al. [51], who showed that the XGboost model had the highest discriminant performance while predicting the severity of COVID-19 pneumonia. Therefore, the key parameters screened by the XGboost model were more accurate than those screened by the RF model. In this study, seven parameters, Mol.Wt, Q^+^, QH^+^, E_LUMO_, minHCsats, MDEC-33, and EG, were identified as the key eigenvalues affecting the harmful outcomes of the ED effects of PAEs. To check the reliability of these seven parameters, we used the data of the seven key characteristic values selected by the XGboost model of the ED effects of PAEs associated with the adverse ending in the AOP as independent variables and the BE data of estrogen and docking of high-molecular-weight PAEs with ED effects in the initial event associated with AOP as dependent variables to construct the XGboost regression model. The code used to construct the model is shown in Appendix A. By evaluating the performance of the model, we found that the R^2^ reached 0.844 (>0.75), and the model qualified [29], but the model R^2^ was lower than that of the XGboost model for the ED effects of PAEs associated with the AOP harmful outcomes. To evaluate the performance and the prediction effect of the model, the mean absolute error (MAE) between the predicted value and the actual value of the model was calculated (4.383), and the average error rate (MAPE) was found to be 5.5%. The results showed that the model had good performance and prediction ability, which confirmed the accuracy and reliability of the seven parameters selected by the XGboost model. In the estrogen effect pathway of PAEs, the molecules of PAEs bind to estrogen receptor proteins to form homodimers and then bind to specific DNA sequences on estrogen DNA response elements [52], which induces the transcription and translation of the subsequent target genes. The effects of the key parameters on the estrogen effect pathway of PAEs were further analyzed. The parameters E_LUMO_, EG, Q^+^, and QH^+^ can affect the hydrogen bonds and hydrophobic interactions during the binding of PAEs and ER, which can either promote or inhibit the binding of the two and affect the binding of normal estrogen and estrogen proteins in the body. This, in turn, can affect the intensity of PAEs to produce ED effects on the female body; these findings confirmed the accuracy and reliability of the model of adverse ending and the credibility of the key influencing factors identified.

### 2.6. Analysis of the Mechanism of the ED Effects of PAEs Associated with AOP Based on the 3D-QSAR Model

The 3D-QSAR model can show the relationship between the structural characteristics of PAEs and their environmental effects. The CoMSIA model constructed in this study had a q^2^ of 0.73 (>0.7) and an n of 10, indicating that the model had good predictive ability. Additionally, the R^2^ was 0.999 (>0.9), and the SEE was 0.007, which indicated that the model had a good fitting ability [53]. From the contour map of the 3D-QSAR model of the ED effects of PAEs (Figure 2, taking the template molecule DAP as an example), we found that the hydrophobic field of PAEs had a contribution rate of 43.6%, the steric field had a contribution rate of 31.8%, and the electrostatic field had a contribution rate of 20.3%, indicating that the hydrophobic field, steric field, and electrostatic field significantly influence the human ED effect.

As shown in Figure 2, in the hydrophobic field, the yellow area appeared near sites 4, 5, 6, 7, and 8 of the common molecular skeletons, indicating that the presence of hydrophobic groups at those sites increased the intensity of the ED effects of PAEs. These results were consistent with those obtained from the sensitivity analysis, where it was shown that the hydrophobic interaction between the ligand and the receptor significantly affects the ED effects of PAEs. Adding hydrophilic groups at the above sites effectively alleviated the ED effects of PAEs. In the stereo field, the green area mainly appeared near sites 4, 5, 7, and 8 of the molecular common skeletons, indicating that increasing the volume of the groups at those sites can increase the ED effects of PAEs. Reducing the groups at those sites can effectively alleviate the ED effects of PAEs. In the electrostatic field, the blue region mainly covered the groups of the common molecular skeleton at sites 6 and 7, and the red region was mainly concentrated near the groups of the common molecular skeleton at sites 5 and 8. The introduction of positive groups in the red region or negative groups in the blue region can effectively alleviate the ED effects of PAEs.

Taking the three-dimensional receptor field of the 3D-QSAR model as an example, reducing the volume of the groups at sites 4, 5, 7, and 8 can effectively alleviate the ED effect of PAEs. The risk effect values of the DEHP, DHP, DBP, and DAP molecules on the harmful outcome path of ED effects were 0.579, 0.517, 0.305, and 0.100, respectively (Table 3). The groups on site 7 of the common molecular skeleton were -C_9_H_17_O_2_, -C_7_H_13_O_2_, -C_5_H_9_O_2_, and -C_4_H_5_O_2_, respectively. The volume of these four groups also decreased, indicating that the potential map characteristics of the 3D-QSAR model can reflect the correlation between the molecular structure of PAEs and their ED effects on the body. We also found that the molecular weights of the four alkyl groups decreased, which confirmed the results of the previous sensitivity analysis (the greater the molecular weight of PAEs, the greater the intensity of the ED effects on the body).

### 2.7. Analysis of the Mechanism by Which PAEs Cause ED Effects Associated with the AOP

#### 2.7.1. Determining the Mechanism by Which PAEs Cause ED Effects Associated with AOP Based on the MD and Amino Acid Residue Analyses

The ED effect of PAEs occurs mainly because of the effect of PAEs on the transcription and translation of subsequent DNA elements after binding to receptor proteins. Therefore, the binding ability of hormone receptor proteins (initial events) strongly influences the ED effects of PAEs in the body. In this study, the amino acid residues around the molecular docking sites of PAEs and hormone protein receptors (6CHW, 1T7T, and 1NAX) were analyzed. A map of the amino acid residues of 66 complexes of three hormone protein receptors docked by 22 PAEs showed that all PAEs were only surrounded by amino acids that interact via VDW forces and electrostatic forces. The amino acids that interact via van der Waals forces were significantly more common than those that interact through electrostatic forces. The results showed that VDW interactions were the dominant factor affecting the binding of PAEs to ER, AR, and THR (6CHW, 1T7T, and 1NAX, respectively).

By analyzing the free energy data accumulation of PAEs and ER, AR, and THR (6CHW, 1T7T, and 1NAX), the affinity dominant energy between them was identified, and the causes of the initial events in AOP were further investigated. In the MD calculation of BE, energy values can be obtained for VDW, electrostatic, polar solvation, and SASA energy [54]. The sum of the four forms the BE data. The results of our analysis show that VDW and polar solvation energy greatly contributed (positively and negatively, respectively) to the BE of PAEs with the three hormone receptor protein complexes. Therefore, the ED effects of PAEs might be weakened by weakening the VDW energy between PAEs and hormone proteins or increasing the polar solvation energy. The effect of VDW energy was greater than that of polar solvation energy, which indicated that van der Waals energy played a key role in the binding of PAEs to the hormone receptor proteins (6CHW, 1T7T, and 1NAX). Therefore, by weakening the VDW energy between PAEs and hormone proteins, the binding ability of the two can be reduced to decrease the ED effects in the body. These results confirm that VDW interactions are the main factor affecting the AOP of the ED effects of PAEs.

Taking BBP (a PAE) as an example, we analyzed the mechanism of the harmful outcome of the ED effect of PAEs. The number of amino acids formed by van der Waals interaction between BBP molecules and the 6CHW, 1T7T, and 1NAX proteins was large (Figure 3a–c), and the VDW energy contribution was the highest in the molecular BE value (Table 4), which matched the results of the above analysis. This indicated that VDW interaction was the main factor affecting the harmful outcome pathway of the ED effects of PAEs.

#### 2.7.2. Analysis of the Mechanism of the ED Effects of PAEs Associated with the AOP Based on Multiple Non-Bonding Forces

In this study, the Discovery Studio 4.0 software was used to identify the non-bonding forces between PAEs and hormone protein receptors 6CHW, 1T7T, and 1NAX-related amino acids (including hydrogen bonding, electrostatic, hydrophobic, halogen, mixed, and unfavorable forces), and the non-bonding forces were calculated using Equations (1)–(3) to locate key amino acids. In the same three-dimensional coordinate system, based on the spatial position coordinates of each atom and amino acid residue and the resultant force mode of the bond length of each non-bonding force, the spatial direction and resultant force of various non-bonding forces can be determined [55]. The relationship between the non-bonding force and the BE was analyzed as shown below.
(1)$$\left\{\begin{array}{c} x=\left(x_{i}-{\;x}_{i}^{\prime }\right)\times [(1/l)/l],i=1,\;2,3\ldots ..n\;\\ y=\left(y_{i}-y_{i}^{\prime }\right)\times [(1/l)/l],i=1,2,3\ldots ..n\\ z=\left(z_{i}-{\;z}_{i}^{\prime }\right)\times [(1/l)/l],i=1,2,3\ldots ..n \end{array}\right.$$
(2)$$\left\{\begin{array}{c} x_{whole}=\sum_{i=1}^{n}x\\ y_{whole}=\sum_{i=1}^{n}y\\ z_{whole}=\sum_{i=1}^{n}z \end{array}\right.$$
(3)$$F=\sqrt[{2}]{{x_{whole}}^{2}+{y_{whole}}^{2}+{z_{whole}}^{2}}$$

Here, $x_{i}$ represents the ith non-bonding force of the atom at the x coordinate in the amino acid residue, $y_{i}$ represents the ith non-bonding force of the atom at the y coordinate in the amino acid residue, $z_{i}$ represents the ith non-bonding force of the atom at the z coordinate in the amino acid residue, $x_{i}^{\prime }$ represents the ith non-bonding force of the atom at the x coordinate in the ligand PAEs, $y_{i}^{\prime }$ represents the ith non-bonding force of the atom at the y coordinate in the ligand PAEs, $z_{i}^{\prime }$ represents the ith non-bonding force of the atom at the z coordinate in the ligand PAEs, l represents the bond length of each non-bonding force, $x_{whole}$ represents the resultant force of all non-bonding forces in the x coordinate of the atom in the amino acid residue, $y_{whole}$ represents the resultant force of all non-bonding forces in the y coordinate of the atom in the amino acid residue, $z_{whole}$ represents the resultant force of all non-bonding forces in the z coordinate of the atom in the amino acid residue, and F represents the resultant force mode of all non-bonding forces.

According to the composition type of non-bonding force, PAEs mainly bind to the estrogen protein 6CHW, the androgen protein 1T7T, and the thyroid hormone protein 1NAX via hydrophobic interactions and hydrogen bonds, but hydrophobic interactions contribute more. As shown in Table 5, the non-bonding forces between PAEs and the estrogen protein 6CHW, the androgen protein 1T7T, and the thyroid hormone protein 1NAX were analyzed. The key hydrophobic amino acid affecting the binding of PAEs to the estrogen protein 6CHW was leucine, and the key amino acid involved in hydrogen bonding was asparagine. The key hydrophobic amino acids that affected the binding of PAEs to the androgen protein 1T7T were tryptophan and valine, and the amino acid involved in hydrogen bonding was arginine. The key hydrophobic amino acids affecting the binding of PAEs to the thyroid hormone protein 1NAX were leucine and isoleucine, and the amino acid involved in hydrogen bonding was asparagine. The PAEs showed a strong binding ability by forming hydrophobic and hydrogen bonding interactions with the identified key amino acids. In this study, based on the sensitivity analysis, the key parameters affecting the initiation event in the AOP associated with the ED effects of PAEs were identified. Molecular weight, E_LUMO_, and QH^+^ were found to affect the intensity of hydrogen bonding and hydrophobic interactions between PAEs and the hormone proteins, which confirmed the accuracy and reliability of the key parameters determined in this study. We also used Equations (18)–(20) to calculate the non-bonding force modes of the six complexes with the largest and smallest BE between PAEs and the estrogen protein 6CHW, the androgen protein 1T7T, and the thyroid hormone protein 1NAX (as shown in Table 6). The results showed that, among the three ED effects, the hydrogen bond force mode and hydrophobic force mode in the complexes with larger ED effects (smaller BE value) of PAEs were significantly higher than those with smaller ED effects (larger BE value). To summarize, PAEs showed a strong binding ability by forming hydrophobic interactions and hydrogen bond interactions with certain key amino acids in the hormone receptor proteins, which was consistent with the results that hydrogen bonds and hydrophobic interactions identified in this study based on sensitivity analysis can significantly affect the occurrence of harmful outcome pathways of the ED effects of PAEs. We also found that the binding ability of PAEs to ER, AR, and THR was mainly affected by hydrophobic interactions and hydrogen bond interactions.

## 3. Methodology

### 3.1. Data Sources

In this study, 22 common PAEs were selected as ligand molecules, and the estrogen receptor (ER) (6CHW), the androgen receptor (AR) (1T7T), and the thyroid hormone receptor (THR) (1NAX) were selected as target proteins to determine the ED effects of PAEs and their mechanism of action. The PAEs were virtually constructed using the ChemDraw 20.0 software, and the contour maps of the three receptor proteins and their corresponding DNA response elements (Figure 4) were obtained from the PDB (Protein Data Bank) [56].

### 3.2. Calculation of the BE between PAEs and Hormone Receptor Proteins Based on Molecular Docking and MD Simulation Methods

In this study, the molecular docking method was used to dock 22 PAEs with the ER, AR, and THR proteins to obtain 66 complexes. Then, MD simulation was performed to calculate the BE of the complexes and the hormonal effects of the 22 PAEs on the human body were determined. Molecular docking can bind ligand and receptor proteins as a complex, which can provide preliminary information to perform MD simulations [57,58]. MD simulation can be performed to simulate the physical trajectory and the state of atoms and molecules based on the principle of Newtonian mechanics [59]. In this study, 22 PAEs (BBP, DIHXP, DAP, DPP, DBP, DHP, DEP, DMP, DIDP, DIHP, DNOP, DEHP, DINP, DMEP, DIOP, DIPP, DIPRP, DNP, DPRP, DIBP, DTDP, and DUP) were introduced into the Discovery Studio 4.0 software as ligand molecules, and the estrogen protein (PDB ID: 6CHW), androgen protein (PDB ID: 1T7T), and thyroid hormone protein (PDB ID: 1NAX) were introduced as receptor proteins for docking. After docking, the structure of the complex was uploaded to the ATB database for data extraction. Finally, using the GROMACS software, the MD simulation of the complex was performed [60,61], the BE of the complex was calculated [62], and the BE value was used to characterize the interference intensity of PAEs on the hormone receptor proteins [25].

### 3.3. Estimation of the Comprehensive Effect Value of the Molecular Initial Events in the AOP Associated with the ED Effects of PAEs Based on the Variation Coefficient Method

The coefficient of variation (CV) is often used to measure the difference in the data, and it is weighted based on the degree of variation of each index [63]; the greater the degree of variation, the greater the weight of the index [64]. This method takes the proportion of the CV of each group of data in the sum of the coefficients of variation of the whole group of data as its weight, avoiding the influence of different dimensions and orders of magnitude of the index [65]. In this study, the data of the three kinds of BE were used as the data source, and the CV method was used to convert them into the combined BE values of 22 PAEs that contained information on the estrogen effect, androgen effect, and thyroid hormone effect of PAEs. The steps and formulae used for the calculations are as follows:(1)First, the original data matrix of BE of PAEs with three hormone protein complexes was established:
(4)$$X=\left[\begin{array}{ccc} X_{11} & X_{12} & X_{13}\\ \vdots & \vdots & \vdots \\ X_{n1} & X_{n2} & X_{n3} \end{array}\right]$$

Here, X represents the BE of the docking of PAEs with hormone proteins, n represents the number of PAEs (n=22).

(2)The standard deviations of three groups of BE data were calculated as follows:


(5)
$$S_{j}=\sqrt{\frac{\sum_{i=1}^{n}\left(X_{ij}-\bar{X_{j}}\right)²}{n}}$$


Here, $S_{j}$ represents the standard deviation of the BE of group j.

(3)The CV of the three groups of BE data was calculated as follows:


(6)
$$V_{j}=C\cdot V=S_{j}/\bar{X_{j}}$$


Here, $V_{j}$ denotes the CV of the BE of group j.

(4)The weights of three groups of BE data were calculated using Equation (7).


(7)
$$W_{j}=V_{j}/\sum_{j=1}^{3}V_{j}$$


Here, $W_{j}$ denotes the weight of the BE of the *j*th group.

(5)After calculating the weights of the three groups of data, the comprehensive effect value of the PAE hormones was calculated using Equation (8).


(8)
$$Y_{i}=X_{i1}\times W_{1}+X_{i2}\times W_{2}+X_{i3}\times W_{3}$$


Here, $W_{1}$, $W_{2}$, and $W_{3}$ represent the weight of the BE for the binding of PAEs to estrogen protein, androgen protein, and thyroid hormone protein, respectively. $Y_{i}$ represents the comprehensive effect value for the binding of the *i*th PAE to the three hormone proteins.

### 3.4. Estimation of the Comprehensive Effect Value of the Key Events in the AOP Associated with the ED Effects of PAEs Based on the Complex Correlation Coefficient Method

A multiple correlation coefficient reflects the degree of correlation between a variable and other independent variables [66], and the weight distribution is based on the independence of each index [67]. The greater the independence, the greater the weight, and vice versa [68]. In this study, the scores of three DNA response elements docked by PAEs and hormone protein complexes were used as the data source, and they were processed using the multiple correlation coefficient method to transform them into DNA response elements for effectively characterizing the ED effects on estrogen, ED effects on androgen, and ED effects on thyroid of 22 PAEs. The specific calculation steps and formulae are as follows:

(1)First, according to Equation (4), the original data matrix of PAE–protein complex docking, along with the original score for its corresponding DNA reaction, was established.

Here, X represents the score of the PAEs–hormone receptor complex binding to the corresponding DNA-reactive element; n represents the number of PAEs (n=22).

(2)According to Equation (9), the correlation coefficient matrix of the scoring index of three groups of DNA response elements was established:



(9)
$$R=\left[\begin{array}{ccc} r_{11} & r_{12} & r_{13}\\ r_{21} & r_{22} & r_{23}\\ r_{31} & r_{32} & r_{33} \end{array}\right]$$



(3)According to Equation (10), multiple correlation coefficients between the scoring data of each group of PAEs and the scoring data of the other two groups were calculated:



(10)
$$R=\left[\begin{array}{cc} R_{m-1} & r_{m}\\ \overset{´}{r_{m}} & 1 \end{array}\right]$$



Here, $R_{m-1}$ represents the correlation coefficient matrix of the scores of other m−1 PAEs, and $r_{m}$ represents the m−1 column vector.

(4)According to Equation (11), the reciprocal of the multiple correlation coefficient of the scoring data of the three groups of PAEs was calculated and normalized to obtain the weight of each group of scoring data:



(11)
$$W_{j}=\frac{1}{\rho_{j}}/\sum_{j=1}^{m}\frac{1}{\rho_{j}}$$



Here, $W_{j}$ represents the weight of the scoring data of PAEs in group *j*; $\rho_{j}$ represents the multiple correlation coefficient of the scoring data of PAEs in group j.

(5)After calculating the weights of the three groups of data, the comprehensive score of the DNA response elements of the PAE–hormone protein complexes was calculated using Equation (12).



(12)
$${Y_{i}=X}_{ia}\times W_{1}+X_{ib}\times W_{2}+X_{ic}\times W_{3}$$



Here, $W_{1}$, $W_{2}$, and $W_{3}$ represent the weights of the scoring data of the docking of PAEs and hormone protein complexes with estrogen DNA response elements, androgen DNA response elements, and thyroid hormone DNA response elements, respectively; $Y_{i}$ represents the comprehensive scoring data of the docking of the *i*th PAE with the hormone protein complex to three hormone DNA response elements.

### 3.5. Estimation of the Comprehensive Effect Value of Adverse Ending in the AOP Associated with the ED Effects of PAEs Based on the Forward Method and the Entropy Method

The entropy method is often used to determine the degree of dispersion of a certain index [52]. In this study, the comprehensive BE between PAEs and proteins and the comprehensive score of their corresponding reaction elements were normalized using the forward method [69]:(13)$$X_{j}^{\prime }=\frac{x_{j-x_{min}}}{x_{max}-x_{min}}$$

Here, $X_{j}^{\prime }$ represents the comprehensive value of BE or the score after normalization of the *j*th PAE; $x_{j}$ represents the comprehensive value of the BE or scoring value of the *j*th PAE; $x_{min}$ is the minimum value of the comprehensive BE or the minimum value of the comprehensive score of all PAEs; $x_{max}$ is the maximum comprehensive BE or the maximum comprehensive score of all PAEs.

The entropy method was used to obtain the weights of the MIEs, KEs, and AEs associated with the ED effects of PAEs for estimating the risk effect value of the harmful outcome pathway. The specific calculation steps and equations are as follows:

(1)First, the entropy values of the two sets of data of comprehensive BE and comprehensive scoring were calculated using Equations (14) and (15):



(14)
$$H_{j}=-k\sum_{i=1}^{n}a_{ij}lna_{ij},i=1,2,\ldots n,\;j=1,2,\ldots m$$


(15)
K=1/ln(n)



Here, $a_{ij}$ represents the comprehensive BE and comprehensive scoring data after data normalization, and $H_{j}$ represents the entropy value of comprehensive BE and comprehensive scoring data.

(2)The weight of the initial event and the key event of the harmful outcome path associated with the ED effects of PAEs was calculated using Equation (16):



(16)
$$W_{j}=\frac{1-H_{j}}{m-\sum_{j=1}^{m}H_{j}}$$



Here, $W_{j}$ represents the respective weight of the comprehensive BE of PAEs and the comprehensive scoring data.

(3)The risk effect value of the harmful outcome related to the molecular ED effects of PAEs was calculated using Equation (17):



(17)
$$Y_{i}=a_{i1}\times W_{1}+a_{i2}\times W_{2}$$



Here, $W_{1}$ and $W_{2}$ represent the weight of the initial event and key event data of the harmful outcome pathway associated with PAEs, respectively, and $Y_{i}$ represents the risk effect value of the harmful outcome pathway associated with the *i*th PAE.

### 3.6. Calculation and Screening of the Molecular Characteristic Values of PAEs

#### 3.6.1. Calculation of the Molecular Characteristic Values of PAEs Using the Gaussian, PaDEL-Descriptor, and ChemDraw Software Programs

In this study, the PaDEL-descriptor software developed by Professor Chun of the National University of Singapore, the ChemDraw software developed by PerkinElmer, and the Gaussian software developed by the Nobel Prize winner John. A. Pople were used to calculate 946 molecular eigenvalues of 22 PAEs. Among them, the PaDEL-Descriptor software was used to calculate some topological and geometric parameters, such as van der Waals volume and atomic number [70]. The ChemDraw 20.0 software was used to calculate some structural parameters and topological parameters of PAEs, such as stereo effect parameters and molecular weight [71]. The Gaussian software was used to calculate some geometric and electronic parameters of PAEs, such as dipole moment and the most positive hydrogen ion charge number [72].

#### 3.6.2. Dimensionality Reduction of the Eigenvalues of PAEs Using Pearson’s Correlation Coefficient (PCC)

We used the PCC method to eliminate irrelevant and highly correlated feature values of PAEs and retain the main features for model construction [73] to reduce the feature value dimension while avoiding the problems of overfitting and low training efficiency [74]. The PCC is defined as the quotient of covariance and standard deviation between two variables [75] and can be calculated as follows:(18)$$\rho (X,\;Y)=\frac{cov(X,\;Y)}{\sigma X\cdot \sigma Y}=\frac{E(XY)-E(X)E(Y)}{\sqrt{E(X^{2})-E^{2}(X)}\sqrt{E(Y^{2})-E^{2}(Y)}}$$

Here, E represents the mathematical expectation, cov (X, Y) represents the covariance of the 2-eigenvalue data of PAEs, σX and σY represent the standard deviations of the eigenvalue data of PAEs *X* and *Y*, respectively. The PCC can range between –1 and 1; its absolute value can be interpreted as follows: >0.8: very strong, 0.6–0.8: strong, 0.4–0.6: medium, 0.2–0.4: weak, and <0.2: very weak [76].

It is easier to implement arbitrary modifications, customizations, and analysis process automation in the PCC model using Python compared to using most other software [77]. Therefore, we used the Python software to independently write a code package for analyzing the characteristic parameters of 946 PAEs, construct a PCC model, and classify and screen according to the classified threshold and Pearson’s matrix, to quickly perform the PCC analysis of the data on the characteristic value of PAEs. The code used to construct the PCC model in this study is shown in Appendix A.

### 3.7. Construction of the Machine Learning Model for the AOP Associated with the ED Effects of PAEs

#### 3.7.1. Construction of the Machine Learning Model for the AOP Associated with the ED Effects of PAEs Based on the MLR Algorithm

The type of linear model constructed to perform the regression learning task is called a linear regression model; it can explain the relationship between dependent and independent variables [78]. When analyzing a multi-factor model, using the linear regression model is simpler [79], and the predicted results are better [80]. A linear regression model can be either univariate or multivariate. Since we investigated the influence of multiple factors on the dependent variables in this study, and the 2D-QSAR model uses the linear regression model, we constructed a multivariate linear regression model and adjusted the internal parameters such as fit_intercept to evaluate the influence of each eigenvalue of the PAEs on the ED effects. The code used to construct the MLR model in this study is shown in Appendix A.

#### 3.7.2. Construction of the Machine Learning Model for the AOP Associated with the ED Effects of PAEs Based on the RF Algorithm

A multivariate linear regression model has several limitations, and the RF algorithm uses an integrated algorithm with high accuracy. Therefore, in this study, we used the RF algorithm to construct a machine-learning model. The RF regression algorithm is a combination algorithm that combines multiple decision trees to reduce the risk of model overfitting. It uses the decision tree as the basic trainer, implements the Bagging integration method, and introduces random attribute selection [81]. Its advantages include easy implementation and fast training [82]. Therefore, we constructed an RF regression model to adjust the internal parameters, such as n_estimators, max_samples, and max_depth. The code used to construct the RF regression model is shown in Appendix A.

#### 3.7.3. Construction of the Machine Learning Model for the AOP Associated with the ED Effects of PAEs Based on the XGboost Algorithm

To avoid the problems of poor performance and poor prediction ability of the RF model in this study, we used the XGboost algorithm with better performance to construct the model. XGboost is an optimized distributed gradient boosting library [83]. It can avoid overfitting [84], and due to its second-order convergence characteristics, XGboost has higher modeling efficiency [85]. Therefore, in this study, we constructed an XGboost regression model to adjust internal parameters, such as n_estimators, max_depth, eta, and gamma. The code used for constructing the XGboost regression model is shown in Appendix A.

### 3.8. Identification of the Key Parameters of the Machine Learning Model for AOP Associated with the ED Effects of PAEs Based on the Sensitivity Analysis

The sensitivity coefficient (SC) is the ratio of the change rate of the model prediction value to the change rate of the input parameter. By performing sensitivity analysis, the key influencing factors that have a greater effect on the model prediction value can be identified [86]. To determine the critical influencing parameters of the AOP of the ED effects of PAEs, we determined the importance of the parameters of the qualified model for the MIEs, KEs, and AE associated with the ED effects of PAEs by performing sensitivity analysis. The SC is shown in Appendix A. In this study, 19 eigenvalues were increased by 10%, 20%, 30%, 40%, and 50%, respectively. The sensitivity coefficients of each eigenvalue in each model were calculated using Equation (19), and the key eigenvalues of each model of the ED effects of PAEs associated with the harmful outcome path were screened. The parameters with larger absolute values of SC were selected as the key eigenvalues of the ED effects of PAEs.
(19)$${SC}_{i}=({∆Y}_{i}/Y_{i})/({∆X}_{i}/X_{i})$$

Here, ${SC}_{i}$ represents the sensitivity coefficient of the *i*th eigenvalue of the PAEs; ΔXi/Xi represents the change rate of the *i*th eigenvalue of the PAEs, and ΔYi/Yi represents the predicted value change rate of y output in three machine learning models.

### 3.9. Construction of the 3D-QSAR Model for the AOP Associated with the ED Effects of PAEs

In this study, a 3D-QSAR model was constructed and analyzed using the SYBYL-X2.0 software [87]. The 3D-QSAR model of PAEs was constructed by using the structure of 22 PAEs as arguments and the comprehensive value of the adverse ending of PAEs as dependent variables. Among the 22 PAEs, 16 were randomly selected as the training set, and the remaining six molecules were used as the test set. The DAP molecules were used as template molecules, and suitable common molecular skeletons were selected for stacking. The similarity index analysis (CoMSIA) was performed to construct the model [88]. After the CoMSIA field parameters were calculated, to test whether the model had a good predictive ability and fitting ability, we calculated the cross-validation coefficient q^2^, the best principal component n, the non-cross-validation coefficient r^2^, the standard deviation SEE, and the test value F to cross-validate the training set molecules.

## 4. Conclusions

In this study, we constructed the RF model and the XGboost model of the ED effect of PAEs associated with AOP using a machine learning method and harmful outcome path coupling. We also constructed a 3D-QSAR model for the harmful outcome of the ED effects of PAEs. By conducting the sensitivity analysis of the RF model and the XGboost model of the ED effects of PAEs associated with AOP and constricting a contour map of the 3D-QSAR model, we found that the initial event was the key event affecting the AOP of the ED effects of PAEs. Our results also showed that molecular weight was the most important factor affecting the harmful outcome path of the ED effects of PAEs. Based on the MD simulations, amino acid residues, and non-bonding force analysis, we found that hydrophobic interactions and hydrogen bonding interactions produced by the combination of PAEs and hormone proteins were the key factors affecting the initial events in the harmful outcome path of ED. Additionally, the key eigenvalues identified by sensitivity analysis significantly influenced the ED effects of PAEs. Our findings provided new insights into the mechanism of the ED effects of PAEs at the molecular level and theoretical guidance for further research and regulation of the ED effects of pollutants such as PAEs. 

## Figures and Tables

**Figure 1 ijms-24-13548-f001:**
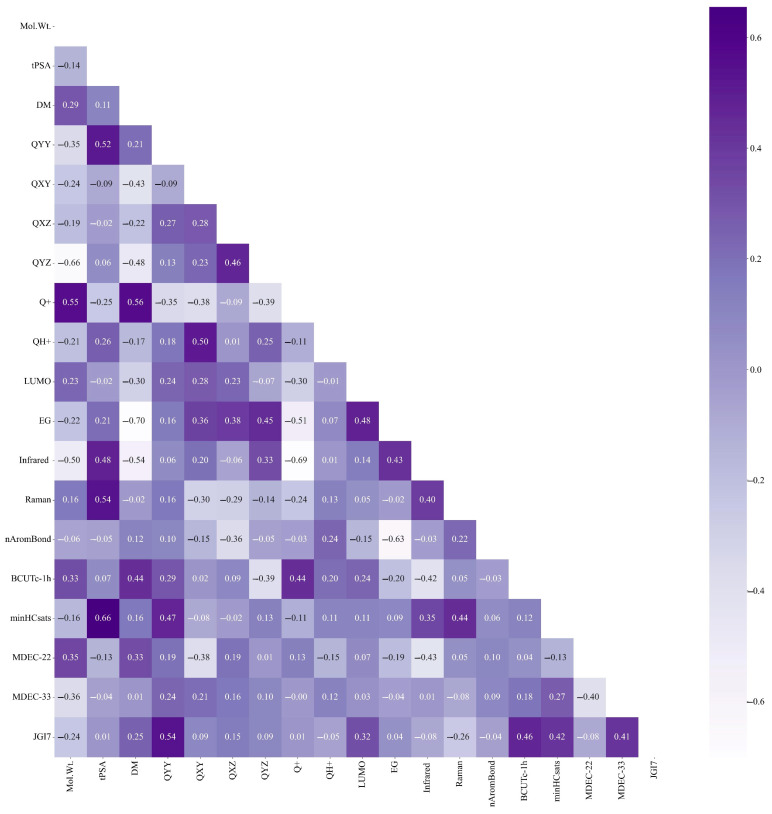
Key parameters of PAEs after dimensionality reduction.

**Figure 2 ijms-24-13548-f002:**
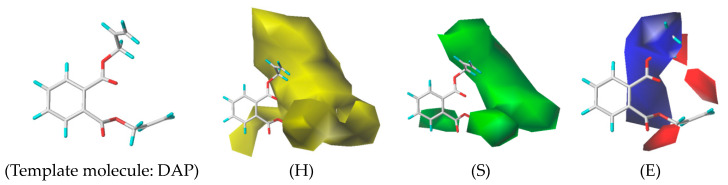
The 3D-QSAR model of ED effects of PAEs on humans (H: Hydrophobic field; S: Steric field; E: Electrostatic field).

**Figure 3 ijms-24-13548-f003:**
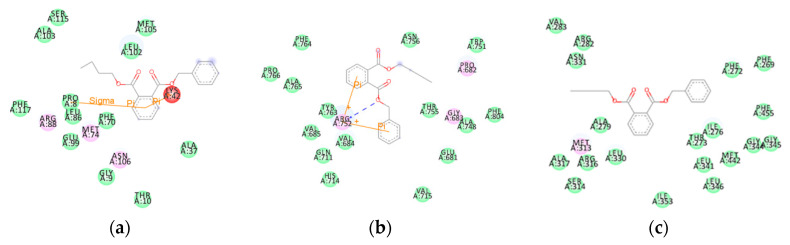
An amino acid residue map of the docking of BBP with three hormone proteins. (**a**) BBP- 6CHW complex, (**b**) BBP-1T7T complex, and (**c**) BBP-1NAX complex. (The purple amino acid residues participated in electrostatic interactions, and the green amino acid residues participated in van der Waals interactions).

**Figure 4 ijms-24-13548-f004:**
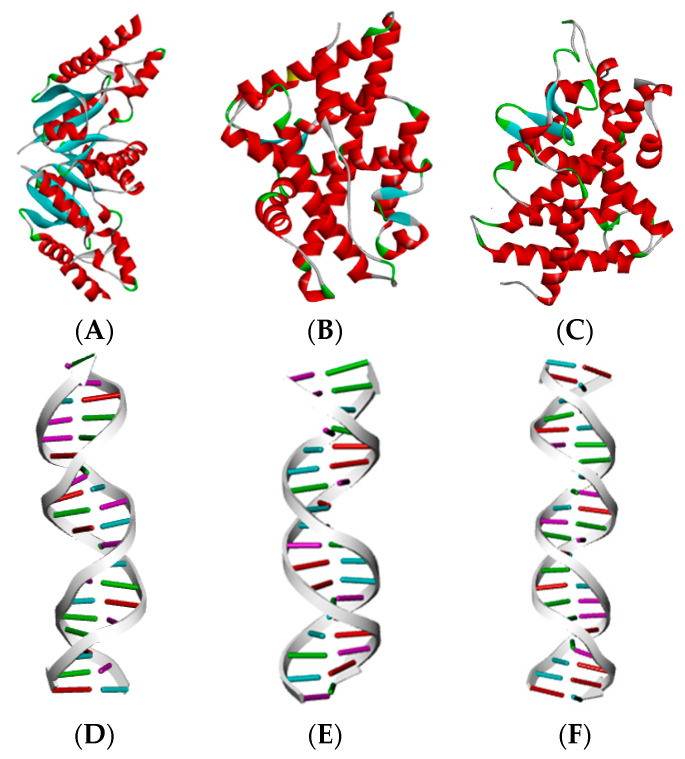
Three ED effector receptor proteins and their corresponding DNA response elements. (**A**–**C**) represent the ER, AR, and THR proteins, respectively; (**D**–**F**) represent the estrogen DNA response element, the androgen DNA response element, and the thyroid hormone DNA response element, respectively.

**Table 1 ijms-24-13548-t001:** BE Values of PAEs to Three Hormone Receptors.

Molecule	3 Hormone Receptor BE (kJ/mol)			Comprehensive BE (kJ/mol)
	Estrogen	Androgen	Thyroid Hormone	
BBP	−70.229	−51.100	−117.692	−78.987
DAP	−40.462	−23.273	−75.020	−45.782
DBP	−54.165	−56.487	−93.811	−67.410
DEHP	−99.145	−93.832	−171.905	−120.350
DEP	−58.338	−24.878	−81.192	−54.698
DHP	−64.202	−62.921	−157.779	−93.273
DIBP	−71.637	−38.937	−100.252	−70.059
DIDP	−80.501	−86.601	−80.479	−82.470
DIHP	−89.240	−76.658	−157.461	−106.660
DIHXP	−118.900	−45.311	−127.952	−97.909
DINP	−87.531	−92.714	−77.541	−86.062
DIOP	−117.551	−80.994	−62.085	−88.229
DIPP	−45.046	−38.138	−120.604	−66.616
DIPRP	−68.632	−24.088	−102.113	−64.750
DMEP	−67.439	−16.720	−86.559	−57.031
DMP	−38.507	−36.674	−67.094	−46.921
DNOP	−64.418	−86.224	−90.899	−79.827
DNP	−85.702	−63.926	−52.928	−68.319
DPP	−89.328	−54.455	−121.823	−88.268
DPRP	−62.242	−30.584	−85.927	−59.448
DTDP	−68.501	−117.851	−85.914	−89.977
DUP	−104.933	−64.944	−100.857	−90.692

**Table 2 ijms-24-13548-t002:** Scoring values of PAEs docking with three hormone DNA responsive elements.

Molecule	3 Hormone DNA Response Element Scores			Comprehensive Score
	Estrogen	Androgen	Thyroid Hormone	
BBP	−271.13	−248.12	−222.50	−238.81
DAP	−282.02	−243.98	−214.83	−236.05
DBP	−292.26	−239.42	−216.21	−238.16
DEHP	−242.50	−250.57	−218.14	−230.44
DEP	−278.80	−249.15	−216.05	−237.10
DHP	−283.64	−241.43	−219.98	−238.78
DIBP	−281.26	−234.38	−216.26	−234.66
DIDP	−277.88	−254.88	−218.08	−239.25
DIHP	−280.27	−242.12	−217.40	−236.71
DIHXP	−261.42	−249.26	−219.13	−234.97
DINP	−272.39	−247.02	−219.25	−237.02
DIOP	−287.61	−254.99	−219.03	−241.99
DIPP	−287.86	−245.44	−222.02	−241.73
DIPRP	−281.06	−250.53	−216.22	−238.00
DMEP	−272.66	−251.07	−214.93	−235.50
DMP	−269.74	−250.82	−217.63	−236.31
DNOP	−278.15	−245.72	−220.25	−238.61
DNP	−282.78	−248.24	−212.29	−235.68
DPP	−272.65	−242.00	−218.67	−235.70
DPRP	−285.74	−240.90	−216.39	−237.11
DTDP	−283.34	−255.34	−225.20	−244.59
DUP	−278.80	−246.96	−246.92	−254.08

**Table 3 ijms-24-13548-t003:** The AOP harmful outcome effect value associated with the ED effects of PAEs.

Molecule	Normalized Value		Harmful Outcome Effect Value
	Comprehensive BE	Comprehensive Score	
BBP	0.445	0.354	0.407
DAP	0.000	0.237	0.100
DBP	0.290	0.327	0.305
DEHP	1.000	0.000	0.579
DEP	0.120	0.282	0.188
DHP	0.637	0.353	0.517
DIBP	0.326	0.179	0.264
DIDP	0.492	0.373	0.442
DIHP	0.816	0.265	0.584
DIHXP	0.699	0.192	0.485
DINP	0.540	0.279	0.430
DIOP	0.569	0.489	0.535
DIPP	0.279	0.477	0.363
DIPRP	0.254	0.320	0.282
DMEP	0.151	0.214	0.177
DMP	0.015	0.249	0.113
DNOP	0.457	0.345	0.410
DNP	0.302	0.222	0.268
DPP	0.570	0.222	0.424
DPRP	0.183	0.282	0.225
DTDP	0.593	0.599	0.595
DUP	0.602	1.000	0.770

**Table 4 ijms-24-13548-t004:** The BE composition of the three hormone proteins after docking with BBP.

Molecular Protein Complex	Combined Free Energy Composition			
	Van der Waals Energy (kJ/mol)	Electrostatic Energy (kJ/mol)	Polar Solvation Energy (kJ/mol)	SASA Energy (kJ/mol)
BBP and 6CHW	−135.950	−32.393	115.064	−16.951
BBP and 1T7T	−155.165	−13.911	134.881	−16.904
BBP and 1NAX	−173.957	−4.647	79.874	−18.963

**Table 5 ijms-24-13548-t005:** Number of hydrophobic amino acids involved in non-bonding interactions between PAEs and hormone proteins.

Amino Acid Residue	Frequency of Occurrence in Hormone Proteins		
	Estrogen Protein	Androgen Protein	Thyroid Hormone Protein
Leucine	39	16	42
Methionine	18	2	16
Alanine	24	14	31
Phenylalanine	9	10	18
Arginine	17	23	25
Proline	14	38	4
Histidine	11	1	3
Lysine	6	7	1
Tyrosine	2	7	0
Valine	1	55	7
Glycine	1	0	0
Tryptophan	0	47	1
Isoleucine	0	1	42
Cysteine	0	0	4

**Table 6 ijms-24-13548-t006:** Non-bonding force mode and BE of PAEs and the hormone protein complex.

ED Effects	Complex	Non-Bond Force Resultant Mode		BE (kJ/mol)
		Hydrogen Bonding Interaction	Hydrophilic Force	
Estrogenic effect	Dmp	1.015028	0.558063	−38.507
	Dihxp	0.360728	0.499671	−118.900
Androgenic effect	Dmep	0.656629	1.443051	−16.720
	Dtdp	0.414266	0.438832	−117.851
Thyroid hormone effect	Dnp	0.748794	0.343049	−52.928
	Dehp	0.399914	0.217793	−171.905

## Data Availability

Not applicable.

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
