# Peer review of "Human Endocrine-Disrupting Effects of Phthalate Esters through Adverse Outcome Pathways: A Comprehensive Mechanism Analysis"

_ijms, 2023, doi:10.3390/ijms241713548_

Round 1

Reviewer 1 Report

The article results in a very interesting topic; phthalates are an issue related to plastics of a concern given the impact on the environment and human health. The authors have covered the topic well and clearly reported the simulations and data obtained. The references are all recent and overall, the article is well structured.

Reviewer 2 Report

The manuscript ijms-2526074 reports theoretical data on the effects of PAEs on human health deduced by Adverse Outcome Pathways.

The topic should be extremely interesting, but simply approached on a mathematical basis without any experimental support makes no sense. Humans (and in general animals) are not a simple organized set of molecules, and the interactions between them and the chemical compounds that surround them are extremely complex and variable at the organ, tissue and cell level, even considering the half-life times of these molecules.

Mathematical modelling, however rigorous, can never provide biological information.

I consider that this manuscript is not appropriate for publication in the International Journal of Molecular Sciences, and I suggest the Authors send it to some Mathematical or engineering journal, fields in which they evidently have greater knowledge.

Reviewer 3 Report

The work of Li et a; is relevant and of importance. However, the authors need to mark relevant aspect as this topic is neither novel nor it its first years of discussions.

The abstract is to telegraphic, should be made in a more fluent form and English editing performed by a native is required.

Phthalate esters (PAEs) binding affinity to estrogen receptors is not mentioned, although weakly estrogenic, this aspect has to be mentioned when talking about endocrine disruptors. PAEs binding to thyroid hormone receptors ( TRα1, TRβ1) is also lacking proper data. The Introduction needs a lot more work.
Also, the following aspects need to be mentioned:
In what types of plastics (from plastic types 1-7) are PAEs present.

Line 38: "of six plasticizers" which are those?

Line 40: "included three PAEs" which are those?

For reader's convenience, these need to be stated clearly.

Line 62: "by constructing the AOP of SARS-CoV-2-induced liver injury" not relevant for the topic of the present article.

Lines 68-74 are not relevant

Reviewer 4 Report

The article is devoted to a topical issue related to the study of the effect of phthalic acid esters on the human endocrine system through adverse effects. The authors propose an interesting innovative approach, which consists in using machine learning methods to study the mechanisms that take place in the human endocrine system under the influence of phthalic acid esters. It should be noted that the method is universal and can be used in other theoretical studies on the influence of negative factors on the endocrine system. However, there are a few comments that would help improve this article.

1. The authors indicate, In this study, 22 common PAEs were selected as ligand molecules, and the estrogen 103 receptor (ER) (6CHW), the androgen receptor (AR) (1T7T), and the thyroid hormone re- 104 receptor (THR ) (1NAX) were selected as target proteins to determine the ED effects of PAEs 105 and their mechanism of action. Did the authors model these molecules on their own or did they use known results? Is this data publicly available? What software package was used to model the molecules?

2. The authors used the well-known machine learning boosting method. This method shows good results. Why did the authors use this particular method? Is this the intuition of the researcher or there are other reasons for this. Perhaps it was worth making a comparison with other well-known methods, for example, neural networks. If not, then a reason must be given.

In general, the article is of interest to the scientific community and can be published after minor revision.

Round 2

Reviewer 2 Report

I understand the point of view expressed by the Authors, but it is my opinion that the manuscript is not suitable for publication in IJMS.